# NAS EVALUATION IS FRUSTRATINGLY HARD

**Antoine Yang**[*]
École Polytechnique[†]
France

**Pedro M Esperança**
Huawei Noah's Ark Lab
London, UK

**Fabio Maria Carlucci**
Huawei Noah's Ark Lab
London, UK

## ABSTRACT

Neural Architecture Search (NAS) is an exciting new field which promises to be as much as a game-changer as Convolutional Neural Networks were in 2012. Despite many great works leading to substantial improvements on a variety of tasks, comparison between different methods is still very much an open issue. While most algorithms are tested on the same datasets, there is no shared experimental protocol followed by all. As such, and due to the under-use of ablation studies, there is a lack of clarity regarding *why* certain methods are more effective than others. Our first contribution is a benchmark of 8 NAS methods on 5 datasets. To overcome the hurdle of comparing methods with different search spaces, we propose using a method's relative improvement over the randomly sampled *average architecture*, which effectively removes advantages arising from expertly engineered search spaces or training protocols. Surprisingly, we find that many NAS techniques struggle to significantly beat the average architecture baseline. We perform further experiments with the commonly used DARTS search space in order to understand the contribution of each component in the NAS pipeline. These experiments highlight that: (i) the use of *tricks* in the evaluation protocol has a predominant impact on the reported performance of architectures; (ii) the cell-based search space has a very narrow accuracy range, such that the seed has a considerable impact on architecture rankings; (iii) the hand-designed macro-structure (cells) is more important than the searched micro-structure (operations); and (iv) the depth-gap is a real phenomenon, evidenced by the change in rankings between 8 and 20 cell architectures. To conclude, we suggest best practices, that we hope will prove useful for the community and help mitigate current NAS pitfalls, e.g. difficulties in reproducibility and comparison of search methods. The code used is available at https://github.com/antoyang/NAS-Benchmark.

## 1 INTRODUCTION

As the deep learning revolution helped us move away from hand crafted features (Krizhevsky et al., 2012) and reach new heights (He et al., 2016; Szegedy et al., 2017), so does Neural Architecture Search (NAS) hold the promise of freeing us from *hand-crafted architectures*, which requires tedious and expensive tuning for each new task or dataset. Identifying the optimal architecture is indeed a key pillar of any Automated Machine Learning (AutoML) pipeline. Research in the last two years has proceeded at a rapid pace and many search strategies have been proposed, from Reinforcement Learning (Zoph & Le, 2017; Pham et al., 2018), to Evolutionary Algorithms (Real et al., 2017), to Gradient-based methods (Liu et al., 2019; Liang et al., 2019). Still, it remains unclear which approach and search algorithm is preferable. Typically, methods have been evaluated on accuracy alone, even though accuracy is influenced by many other factors besides the search algorithm. Comparison between published search algorithms for NAS is therefore either very difficult (complex training protocols with no code available) or simply impossible (different search spaces), as previously pointed out (Li & Talwalkar, 2019; Sciuto et al., 2019; Lindauer & Hutter, 2019).

NAS methods have been typically decomposed into three components (Elsken et al., 2019; Li & Talwalkar, 2019): search space, search strategy and model evaluation strategy. This division is

---

[*] antoineyang3@gmail.com, {pedro.esperanca,fabio.maria.carlucci}@huawei.com
[†] This work was done when the first author was an intern at Huawei Noah's Ark Lab, London, United Kingdom.

important to keep in mind, as an improvement in any of these elements will lead to a better final performance. But is a method with a more (manually) tuned search space a better *AutoML* algorithm? If the key idea behind NAS is to find the optimal architecture, without human intervention, why are we devoting so much energy to infuse expert knowledge into the pipeline? Furthermore, the lack of ablation studies in most works makes it harder to pinpoint which components are instrumental to the final performance, which can easily lead to *Hypothesizing After the Results are Known* (HARKing; Gencoglu et al., 2019).

Paradoxically, the huge effort invested in finding better search spaces and training protocols, has led to a situation in which *any* randomly sampled architecture performs almost as well as those obtained by the search strategies. Our findings suggest that most of the gains in accuracy in recent contributions to NAS have come from manual improvements in the training protocol, not in the search algorithms.

As a step towards understanding which methods are more effective, we have collected code for 8 reasonably fast (search time of less than 4 days) NAS algorithms, and benchmarked them on 5 well known CV datasets. Using a simple metric—the relative improvement over the average architecture of the search space—we find that most NAS methods perform very similarly and rarely substantially above this baseline. The methods used are DARTS, StacNAS, PDARTS, MANAS, CNAS, NSGANET, ENAS and NAO. The datasets used are CIFAR10, CIFAR100, SPORT8, MIT67 and FLOWERS102.

Through a number of additional experiments on the widely used DARTS search space (Liu et al., 2019), we will show that: **(a)** how you train your model has a much bigger impact than the actual architecture chosen; **(b)** different architectures from the same search space perform very similarly, so much so that **(c)** hyperparameters, like the number of cells, or the seed itself have a very significant effect on the ranking; and **(d)** the specific operations themselves have less impact on the final accuracy than the hand-designed macro-structure of the network. Notably, we find that the 200+ architectures sampled from this search space (available from the link in the abstract) are all within a range of one percentage point (top-1 accuracy) after a standard full training on CIFAR10. Finally, we include some observations on how to foster reproducibility and a discussion on how to potentially avoid some of the encountered pitfalls.

## 2 RELATED WORK

As mentioned, NAS methods have the potential to truly revolutionize the field, but to do so it is crucial that future research avoids common mistakes. Some of these concerns have been recently raised by the community.

For example, Li & Talwalkar (2019) highlight that most NAS methods a) fail to compare against an adequate baseline, such as a properly implemented random search strategy, b) are overly complex, with no ablation to properly assign credit to the important components, and c) fail to provide all details needed for successfully reproducing their results. In our paper we go one step further and argue that the relative improvement over the average (randomly sampled) architecture is an useful tool to quantify the effectiveness of a proposed solution and compare it with competing methods. To partly answer their second point, and understand how much the final accuracy depends on the specific architecture, we implement an in-depth study of the widely employed DARTS (Liu et al., 2019) search space and perform an ablation on the commonly used training techniques (e.g. Cutout, DropPath, AutoAugment).

In addition, Sciuto et al. (2019) also took the important step of systematically using fair baselines, and suggest random search with early stopping, averaged over multiple seeds, as an extremely competitive baseline. They find that the search spaces of three methods investigated (DARTS, ENAS, NAO) have been expertly engineered to the extent that any randomly selected architecture performs very well. In contrast, we show that even random sampling (without search) provides an incredibly competitive baseline. Our relative improvement metric allows us to isolate the contribution of the search strategy from the effects of the search space and training pipeline. Thus, we further confirm the authors' claim, showing that indeed the average architecture performs extremely well and that how you train a model has more impact than any specific architecture.

## 3 NAS BENCHMARK

In this section we present a systematic evaluation of 8 methods on 5 datasets using a strategy that is designed to reveal the quality of each method's search strategy, removing the effect of the manually-engineered training protocol and search space. The goal is to find general trends and highlight common features rather than just pin-pointing the most accurate algorithm.

Understanding why methods are effective is not an easy task: most introduce variations to previous search spaces, search strategies, and training protocols—with ablations disentangling the contribution of each component often incomplete or missing. In other words, how can we be sure that a new state-of-the-art method is not so simply due to a better engineered search space or training protocol? To address this issue we compare a set of 8 methods with randomly sampled architectures from their respective search spaces, and trained with the same protocol as the searched architectures.

The ultimate goal behind NAS should be to return the optimal model for *any* dataset given, at least within the limits of a certain task, and we feel that the current practices of searching almost exclusively on CIFAR10 go against this principle. Indeed, to avoid the very concrete risk of overfitting to this set of data, NAS methods should be tested on a variety of tasks. For this reason we run experiments on 5 different datasets.

### 3.1 METHODOLOGY

**Criteria for dataset selection.** We selected datasets to cover a variety of subtasks within image classification. In addition to the standard **CIFAR10** we select **CIFAR100** for a more challenging object classification problem (Krizhevsky, 2009); **SPORT8** for action classification (Li & Fei-Fei, 2007); **MIT67** for scene classification (Quattoni & Torralba, 2009); and **FLOWERS102** for fine-grained object classification (Nilsback & Zisserman, 2008). More details are given in the Appendix.

**Criteria for method selection.** We selected methods which (a) have open-source code, or provided it upon request, and (b) have a reasonable running time, specifically a search time under 4 GPU-days on CIFAR10. The selected methods are: **DARTS** (Liu et al., 2019), **StacNAS** (Li et al., 2019), **PDARTS** (Xin Chen, 2019), **MANAS** (Carlucci et al., 2019), **CNAS** (Weng et al., 2019), **NSGANET** (Lu et al., 2018), **ENAS** (Pham et al., 2018), and **NAO** (Luo et al., 2018). With the exception of NAO and NSGANET, all methods are DARTS variants and use weight sharing.

**Evaluation protocol.** NAS algorithms usually consist of two phases: (i) *search*, producing the best architecture according to the search algorithm used; (ii) *augmentation*, consisting in training from scratch the best model found in the search phase. We evaluate methods as follows: 1. Sample 8 architectures from the search space, uniformly at random, and use the method's code to augment these architectures (same augment seed for all); 2. Use the method's code to search for 8 architectures and augment them (different search seed, same augment seed); 3. Report mean and standard deviation of the top-1 test accuracy, obtained at the end of the augmentation, for both the randomly sampled and the searched architectures;

Since both learned and randomly sampled architectures share the same search space and training protocol, calculating a relative improvement over this random baseline as $RI = 100 \times (Acc_m - Acc_r)/Acc_r$ can offer insights into the quality of the search strategy alone. $Acc_m$ and $Acc_r$ represent the top-1 accuracy of the search method and random sampling strategies, respectively. A good, general-purpose NAS method is expected to yield $RI > 0$ consistently over different searches and across different subtasks. We emphasize that the comparison is not against random search, but rather against random sampling, i.e., the *average* architecture of the search space. For example, in the DARTS search space, for each edge in the graph that defines a cell we select one out of eight possible operations (e.g. pooling or convolutions) with uniform probability $1/8$.

Hyperparameters are optimized on CIFAR10, according to the values reported by the corresponding authors. Since most methods do not include their optimization as part of the search routine, we assumed them to be robust and generalizable to other tasks. As such, aside from scaling down the architecture depending on dataset size, experiments on other datasets use the same hyperparameters. Other training details and references are given in the Appendix.

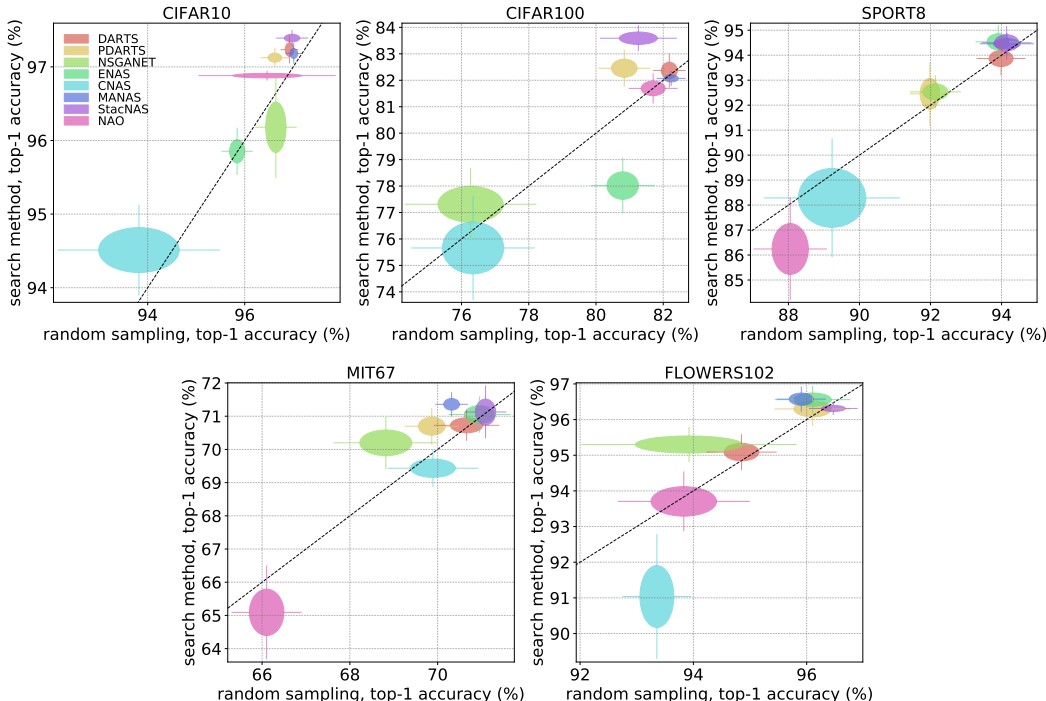

Figure 1: Comparison of search methods and random sampling from their respective search spaces. Methods lying in the diagonal perform the same as the average architecture, while methods above the diagonal outperform it. See also Table 1.

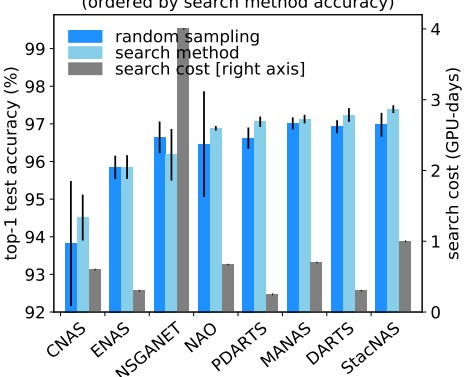

Figure 2: Performance and computational cost of the search phase on CIFAR10.

Table 1: Relative improvement metric, $RI = 100 \times (Acc_m - Acc_r)/Acc_r$ (in %), where $Acc_m$ and $Acc_r$ are the accuracies of the search method and random sampling baseline, respectively.

|  | C10 | C100 | S8 | M67 | F102 |
|---|---|---|---|---|---|
| DARTS | 0.32 | 0.23 | -0.13 | 0.10 | 0.25 |
| PDARTS | 0.52 | 1.20 | 0.51 | 1.19 | 0.20 |
| NSGANET | -0.48 | 1.37 | 0.43 | 2.00 | 1.47 |
| ENAS | 0.01 | -3.44 | 0.67 | 0.13 | 0.47 |
| CNAS | 0.74 | -0.89 | -1.06 | -0.66 | -2.48 |
| MANAS | 0.18 | -0.20 | 0.33 | 1.48 | 0.70 |
| StacNAS | 0.43 | 2.87 | 0.38 | 0.05 | -0.16 |
| NAO | 0.44 | -0.01 | -2.05 | -1.53 | -0.13 |

## 3.2 RESULTS

Figure 1 shows the evaluation results on the 5 datasets, from which we draw two main conclusions. First, the improvements over random sampling tend to be small. In some cases the average performance of a method is even below the average randomly sampled architecture, which suggests that the search methods are not converging to desirable architectures. Second, the small range of accuracies obtained hints at narrow search spaces, where even the worst architectures perform reasonably well. See Section 5 for more experiments corroborating this conclusion.

We observe also that, on CIFAR10, the top half of best-performing methods (PDARTS, MANAS, DARTS, StacNAS) all perform similarly and positively in relation to their respective search spaces, but more variance is seen on the other datasets. This could be explained by the fact that most methods' hyperparameters have been optimized on CIFAR10 and might not generalize as well on

different datasets. As a matter of fact, we found that all NAS methods neglect to report the time needed to optimize hyperparameters. In addition, Table 1 shows the relative improvement metric $RI$ (see intro to Section 3) for each method and dataset.

The computational cost of searching for architectures is a limiting factor in their applicability and, therefore, an important variable in the evaluation of NAS algorithms. Figure 2 shows the performance as well and the computational cost of the search phase on CIFAR10.

## 4 COMPARISON OF TRAINING PROTOCOLS

In this section we attempt to shed some light on the surprising results of the previous section. We noticed that there was a much larger differences between the random baselines of different methods than the actual increase in performance of each approach. We hypothesized that *how* a network is trained (the training protocol) has a larger impact on the final accuracy than *which* architecture is trained, for each search space. To test this, we performed sensitivity analysis using the most common performance-boosting training protocols.

### 4.1 METHODOLOGY

We decided to evaluate architectures from the commonly used DARTS search space (Liu et al., 2019) on the CIFAR10 dataset. We use the following process: 1) sample 8 random architectures, 2) train them with different training protocols (details below) and 3) report mean, standard deviation and maximum of the top-1 test accuracy at the end of the training process.

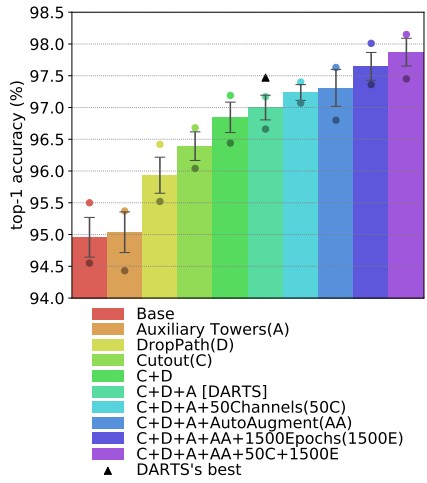

Figure 3: Comparison of different augmentation protocols for the DARTS search space on CIFAR10. Same colored dots represent minimum and maximum accuracies in the 8 runs.

**Training protocols.** The simplest training protocol, which we will call **Base** is similar to the one used in DARTS, but with all tricks disabled: the model is simply trained for 600 epochs. On the other extreme, our full protocol uses several tricks which have been used in recent works (Xie et al., 2019b; Nayman et al., 2019): Auxiliary Towers (**A**), DropPath (**D**; Larsson & Shakhnarovich, 2017), Cutout (**C**; DeVries & Taylor, 2017), AutoAugment (**AA**; Cubuk et al., 2018), extended training for 1500 epochs (**1500E**), and increased number of channels (**50C**). In between these two extremes, by selectively enabling and disabling each component, we evaluated a further 8 intermediate training protocols. When active, DropPath probability is 0.2, cutout length is 16, auxiliary tower weight is 0.4, and AutoAugment combined with Cutout are used after standard data pre-processing techniques previously described, as in Popien (2019).

### 4.2 RESULTS

As shown in Figure 3, a large difference of over 3 percentage points (p.p.) exists between the simplest and the most advanced training protocols. Indeed, this is much higher than any improvement over random sampling observed in the previous section: for example, on CIFAR10, the best improvement observed was 0.69 p.p. In other words, the training protocol is often far more important than the architecture used. Note that the best accuracy of the 8 random architectures training with the best protocol is 98.15%, which is only 0.25 p.p. below state-of-the-art (Nayman et al., 2019).

To summarize, it seems that most recent state-of-the-art results, though impressive, cannot always be attributed to superior search strategies. Rather, they are often the result of expert knowledge applied to the evaluation protocol. In Figure 10 (Appendix A.3.1) we show similar results when training a ResNet-50 (He et al., 2016) with the same protocols.

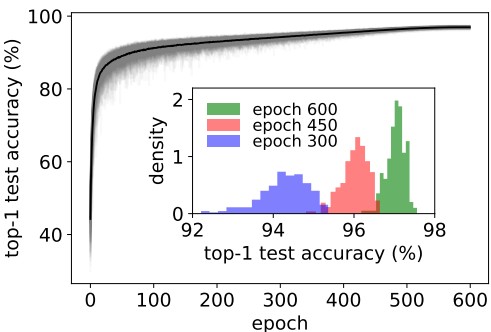 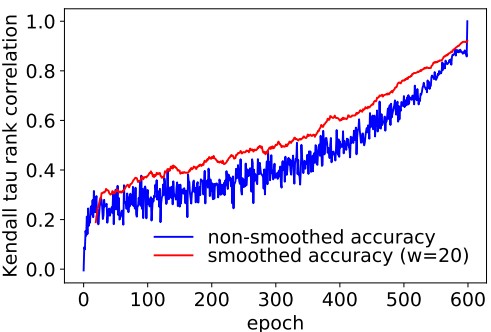

Figure 4: Training curves for the 214 randomly sampled architectures. Inset plot shows the histogram of accuracies at different epochs.

Figure 5: Correlation between accuracies at different epochs and final accuracy, using raw and smoothed accuracies over a window $w$.

## 5 STUDY OF DARTS' SEARCH SPACE

### 5.1 DISTRIBUTION OF THE RANDOM SAMPLING WITHIN DARTS SEARCH SPACE

To better understand the results from the previous section, we sampled a considerable number of architectures (214) from the most commonly used search space (Liu et al., 2019) and fully trained them with the matching training protocol (Cutout+DropPath+Auxiliary Towers). This allows us to get a sense of how much variance exists between the different models (training statistics are made available at the link in the abstract).

As we can observe from Figure 4, architectures sampled from this search space all perform similarly, with a mean of $97.03 \pm 0.23$. The worst architecture we found had an accuracy of $96.18$, while the best achieved $97.56$.

To put this into perspective, many methods *using the same training protocol*, fall within (or very close to) the standard deviation of the average architecture. Furthermore, as we can observe in Figure 7, the number of cells (a human-picked hyperparameter) has a much larger impact on the final accuracy.

In Figure 5 we used the training statistics of the 214 models to plot the correlation between test accuracies at different epochs: it grows slowly in an almost linear fashion. We note that using the moving average of the accuracies yields a stronger correlation, which could be useful for methods using early stopping.

### 5.2 OPERATIONS

To test whether the results from the previous section were due to the choice of available operations, we developed an intentionally sub-optimal search space containing 4 plain convolutions ($1\times1$, $3\times3$, $7 \times 7$, $11 \times 11$), 2 max pooling operators ($3 \times 3$, $5 \times 5$) plus the *none* and *skip connect* operations. This proposed search space is clearly more parameter inefficient compared to the commonly used DARTS one (which uses both dilated and separable ones), and we expect it to perform worse.

We sampled 56 architectures from this new search space and trained them with the DARTS training protocol (Cutout+DropPath+Auxiliary Towers), for fair comparison with the results from the previous section. Figure 6 shows the resulting histogram, together with the one obtained from the classical DARTS space of operations. The two distributions are only shifted by $0.18$ accuracy points. Given the minor difference in performance, the specific operations are not a key ingredient in the success of this search space. Very likely, it's the well engineered cell structure that allows the model to perform as well as it does.

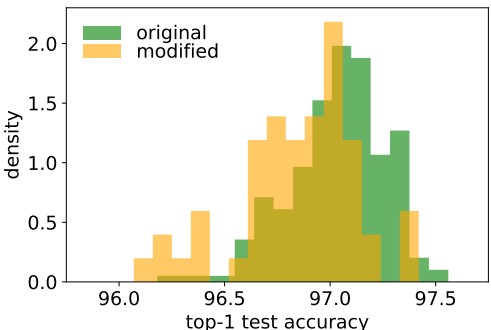

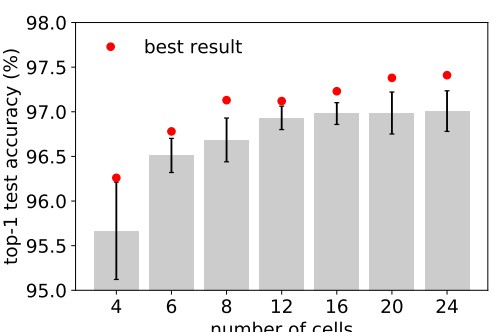

Figure 6: Histograms of the final accuracies (600 epochs) for architectures sampled from the DARTS search space (214 models) and our modified version (56 models).

Figure 7: Performance of 16 randomly sampled architectures with different numbers of cells. Error bars represent standard deviation.

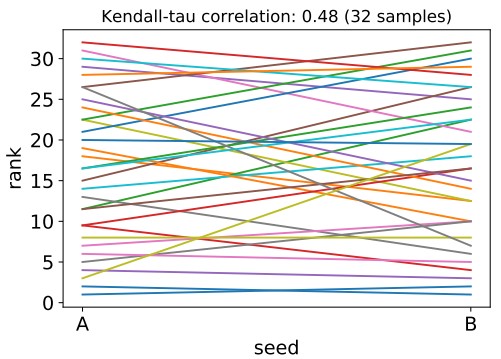

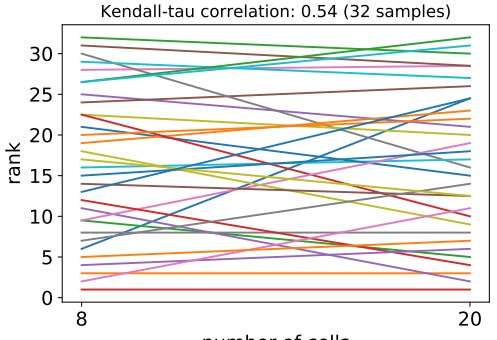

Figure 8: Changes in the ranking of different architectures when trained with two different seeds (A and B).

Figure 9: Changes in the ranking of different architectures when trained with different numbers of cells (same seed).

### 5.3 DOES CHANGING SEED AND NUMBER OF CELLS AFFECT RANKING?

A necessary practice for many weight-sharing methods (Liu et al., 2019; Pham et al., 2018) is to restart the training from scratch after the search phase, with a different number of cells. Recent works have warned that this procedure might negatively affect ranking; similarly, the role of the seed has been previously recognized as as a fundamental element in reproducibility (Li & Talwalkar, 2019; Sciuto et al., 2019).

To test the impact of seed, we randomly sampled 32 architectures and trained them with two different seeds (Figure 8). Ranking is heavily influenced, as the Kendall tau correlation between the two sets of training is $0.48$. On average, the test accuracy changes by $0.13\% \pm 0.08$ (max change is $0.39\%$), which is substantial considering the small gap between random architectures and NAS methods.

To test the depth-gap we trained another 32 with different number of cells (Figure 9). The correlation between the two different depths is not very strong as measured by Kendall Tau ($0.54$), with architectures shifting up and down the rankings by up to $18$ positions (out of 32). Methods employing weight sharing (WS) would see an even more pronounced effect as the architectures normally chosen at $8$ cells would have been training sub-optimally due to the WS itself (Sciuto et al., 2019).

These findings point towards two issues. The first is that since the seed has such a large effect on ranking, it stands to reason that the final accuracy reported should be averaged over multiple seeds. The second is that, if the lottery ticket hypothesis holds—so that specific sub-networks are better mainly due to their lucky initialization; Frankle & Carbin., 2018—together with our findings, this could be an additional reason why methods searching on a different number of cells than the final model, struggle to significantly improve on the average randomly sampled architecture.

## 6    DISCUSSION AND BEST PRACTICES

In this section we offer some suggestions on how to mitigate the issues in NAS research.

**Augmention tricks:**   while achieving higher accuracies is clearly a desirable goal, we have shown in section 4, that using well engineered training protocols can hide the contribution of the search algorithm. We therefore suggest that both results, with and without training tricks, should be reported. An example of best practice is found in Hundt et al. (2019).

**Search Space:**   it is difficult to evaluate the effectiveness of any given proposed method without a measure of how good randomly sampled architectures are. This is not the same thing as performing a random search which is a search strategy in itself; random sampling is simply used to establish how good the average model is. A simple approach to measure the variability of any new given search space could be to randomly sample $k$ architectures and report mean and standard deviation. We hope that future works will attempt to develop more expressive search spaces, capable of producing both good and bad network designs. Restricted search spaces, while guaranteeing good performance and quick results, will inevitably be constrained by the bounds of expert knowledge (local optima) and will be incapable of reaching more truly innovative solutions (closer to the global optima). As our findings in section 5.2 suggest, the overall wiring (the macro-structure) is an extremely influential component in the final performance. As such, future research could investigate the optimal wiring at a global level: an interesting work in this direction is Xie et al. (2019a).

**Multiple datasets:**   as the true goal of AutoML is to minimize the need for human experts, focusing the research efforts on a single dataset will inevitably lead to algorithmic overfitting and/or methods heavily dependent on hyperparameter tuning. The best solution for this is likely to test NAS algorithms on a battery of datasets, with different characteristics: image sizes, number of samples, class granularity and learning task.

**Investigating hidden components:**   as our experiments in Sections 4 and 5.2 show, the DARTS search space is not only effective due to specific operations that are being chosen, but in greater part due to the overall macro-structure and the training protocol used. We suggest that proper ablation studies can lead to better understanding of the contributions of each element of the pipeline.

**The importance of reproducibility:**   reproducibility is of extreme relevance in all sciences. To this end, it is very important that authors release not only their best found architecture but also the corresponding seed (if they did not average over multiple ones), as well as the code and the detailed training protocol (including hyperparameters). To this end, NAS-Bench-101 (Ying et al., 2019), a dataset mapping architectures to their accuracy, can be extremely useful, as it allows the quality of search strategies to be assessed in isolation from other NAS components (e.g. search space, training protocol) in a quick and reproducible fashion. The code for this paper is open-source (link in the abstract). We also open-source the 270 trained architectures used in Section 5.

**Hyperparameter tuning cost:** tuning hyperparameters in NAS is an extremely costly component. Therefore, we argue that either (i) hyperparameters are general enough so that they do not require tuning for further tasks, or (2) the cost is included in the search budget.

## 7    CONCLUSIONS

AutoML, and NAS in particular, have the potential to truly democratize the use of machine learning for all, and could bring forth very notable improvements on a variety of tasks. To truly step forward, a principled approach, with a focus on fairness and reproducibility is needed.

In this paper we have shown that, for many NAS methods, the search space has been engineered such that all architectures perform similarly well and that their relative ranking can easily shift. We have furthermore showed that the training protocol itself has a higher impact on the final accuracy than the actual network. Finally, we have provided some suggestions on how to make future research more robust to these issues.

We hope that our findings will help the community focus their efforts towards a more general approach to automated neural architecture design. Only then can we expect to learn from NAS-generated architectures as opposed to the current paradigm where search spaces are heavily influenced by our current (human) expert knowledge.

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

Table 2: Hyperparameters for different NAS methods. "S" denotes search stage and "A" denotes augmentation stage. For learning rates, $a \downarrow b$ means cosine annealing from $a$ to $b$.

| | Method | | | | |
|---|---|---|---|---|---|
| | DARTS | StacNAS | PDARTS | MANAS | CNAS |
| batch size S | 64 | 64 | 96 | 64 | 64 |
| batch size A | 96 | 96 | 128 | 128 | 64 |
| init channels S | 16 | 16 | 16 | 16 | 16 |
| init channels A | 36 | 36 | 36 | 36 | 36 |
| epochs S | 50 | 100+100 | 25+25+25 | 50 | 60 |
| epochs A | 600 | 600 | 600 | 600 | 200 |
| optimizer S/A | SGD | SGD | SGD | SGD | Adam |
| learning rates S | $.025 \downarrow .001$ | $.025 \downarrow .001$ | $.025 \downarrow .0$ | $.025 \downarrow .001$ | $.025\text{--}.003$ |
| learning rates A | $.025 \downarrow .0$ | $.025 \downarrow .0$ | $.025 \downarrow .0$ | $.025 \downarrow .0$ | $.025\text{--}.001$ |
| weight decay S/A | $3 \times 10^{-4}$ | $3 \times 10^{-4}$ | $3 \times 10^{-4}$ | $3 \times 10^{-4}$ | $3 \times 10^{-4}$ |
| optimizer arch | Adam | Adam | Adam | — | Adam |
| learning rates arch | $3 \times 10^{-4}$ | $3 \times 10^{-4}$ | $6 \times 10^{-4}$ | — | $3 \times 10^{-4}$ |
| weight decay arch | $10^{-3}$ | $10^{-3}$ | $10^{-3}$ | — | $10^{-3}$ |
| nb cells S CIFAR | 8 | 14/20 | 5/11/17 | 8 | 6 |
| nb cells A CIFAR | 20 | 20 | 20 | 20 | 20 |
| nb cells S other datasets | 8 | 8/8 | 8/8/8 | 8 | 6 |
| nb cells A other datasets | 8 | 8 | 8 | 8 | 8 |
| nb intermediate nodes | 4 | 4 | 4 | 4 | 6 |

## A  APPENDIX

This section details the datasets and the hyperparameters used for each method on each dataset. Search spaces were naturally left unchanged. Hyperparameters were chosen as close as possible to the original paper and occasionally updated to more recent implementations. The network size was tuned similarly for all methods for SPORT8, MIT67 and FLOWERS102. All experiments were run on NVIDIA Tesla V100 GPUs.

### A.1  METHODS AND HYPERPARAMETERS

During search, models are trained/validated on the training/validation subsets, respectively During the final evaluation, the model is trained on the training+validation subsets and tested on the test subset.

**Common hyperparameters.** All 8 methods share a common number of hyperparameters precised here. When SGD optimizer is used, momentum is .9 while when Adam is used, momentum is $\beta = (0.5, 0.999)$. Gradient clipping is set at 5.

**DARTS, StacNAS, PDARTS, MANAS, CNAS common hyperparameters.** These methods are inspired by DARTS code-wise and consequently share a common number of hyperparameters, which we precise in table 1.

**DARTS.** We used the following repository : https://github.com/khanrc/pt.darts. It notably updates the official implementation to a pytorch version posterior to 0.4. Additional enhancements include cutout of size 16 (DeVries & Taylor, 2017), path dropout of probability 0.2 (Larsson & Shakhnarovich, 2017), and auxiliary tower with weight 0.4.

**StacNAS.** We used an unofficial implementation provided by the authors. The search process consists of 2 stages, of which the details are given in table 1. Additional enhancements are the same as DARTS.

**PDARTS.** We used the official implementation : https://github.com/chenxin061/pdarts. The search process consists of 3 stages, of which general details are given in table 1. At stage 1, 2 and 3 respectively, the number of operations decreases from 8 to 5 to 3, and the dropout probability on skip-connect increases from 0.0 to 0.4 to 0.7 for CIFAR10, SPORT8, MIT67 and FLOWERS102

(0.1 to 0.2 to 0.3 for CIFAR100). Discovered cells are restricted to keep at most 2 skip-connect operations. Additional enhancements include cutout of size 16 (DeVries & Taylor, 2017), DropPath of probability 0.3 (Larsson & Shakhnarovich, 2017) and auxiliary tower with weight 0.4.

**MANAS.** We used an unofficial implementation provided by the authors. The reward baseline is 5, gamma is 0.1 (0.07 for SPORT8, 0.05 for FLOWERS102, 0.01 for MIT67) and the Boltzmann temperature decays from 1000 to 200 (300 to 100 for SPORT8, 5000 to 2000 for MIT67). Additional enhancements are the same as DARTS.

**CNAS.** We used the official implementation : https://github.com/tianbaochou/CNAS. Label Smoothing is used with epsilon 0.1. Other additional enhancements include cutout of size 16 (De-Vries & Taylor, 2017), DropPath of probability 0.25 (Larsson & Shakhnarovich, 2017).

**NSGANET.** We used the official implementation: https://github.com/ianwhale/nsga-net. The search is done in the micro search space (with 2 cells to search, 9 operations in the search space, 5 blocks in each cell) on 8 layers networks. The population size is 40, the number of generations is 30 and the number of offsprings created by generation is 20. Networks are trained for 20 epochs, with batch size 128, and the initial number of channels is 16. For architecture evaluation, the network is composed of 20 cells for CIFAR10 and CIFAR100, and 8 cells for SPORT8, MIT67 and FLOWERS102. The final selected models are trained for 600 epochs with batch size 96 and the initial number of channels 34. Momentum SGD is used with initial learning rate $\eta_w = 0.025$ (annealed down to zero following a cosine schedule), and weight decay $3 \times 10^{-4}$. The filter increment is set to 4, and squeeze and excitation is used. Additional enhancements include cutout of size 16 (DeVries & Taylor, 2017), DropPath of probability 0.2 (Larsson & Shakhnarovich, 2017) and auxiliary tower with weight 0.4.

**NAO.** We used the official Pytorch implementation: https://github.com/renqianluo/NAO_pytorch. The LSTM model used to encode architecture has a token embedding size of 48 and a hidden state size of 96. The LSTM model used to decode architecture has a hidden state size of 96. The encoder and decoder are trained using Adam for 1000 epochs with a learning rate of 0.001. The trade-off parameters is $\lambda = 0.9$. The step size to perform continuous optimization is $\eta = 10$. The number of nodes is fixed to 5, normal cell is stacked 3 (2 for SPORT8, MIT67 and FLOWERS102) times to form the CNN architecture, which corresponds to a 11 (8 for SPORT8, MIT67 and FLOWERS102) cells network and the initial number of channels is 20. Networks are trained for 100 epochs with batch size 64 for training, and validated for 20 epochs with batch size 500. For architecture evaluation, the final CNN architecture is a 20 cells network (8 cells network). This network is trained for 600 epochs with batch size 128 for the training set (96 for both for SPORT8, MIT67 and FLOW-ERS102), 500 for the validation set (128 for both for SPORT8, MIT67 and FLOWERS102) and the initial number of channels is 36. Momentum SGD is used with initial learning rate $\eta_w = 0.025$ (annealed down to zero following a cosine schedule) and weight decay $3 \times 10^{-4}$. Additional enhancements include cutout of size 16 (DeVries & Taylor, 2017), DropPath of probability 0.2 (Larsson & Shakhnarovich, 2017), dropout of probability 0.4 (Srivastava et al., 2014), and auxiliary tower with weight 0.4.

**ENAS.** We used the official tensorflow implementation for experiments on CIFAR10 and CIFAR100: https://github.com/melodyguan/enas Experiments on SPORT8, MIT67 and FLOWERS102 are done using a more recent version in Pytorch for the search : https://github.com/MengTianjian/enas-pytorch. Because only the search process is implemented there, the evaluation code used is the same as DARTS. During the search, networks are composed of 8 cells. The shared parameters $w$ are trained with Nesterov momentum (Nesterov, 1983), weight decay $10^{-4}$, gradient clipping 5, batch size 160, 20 output filters, and a cosine learning rate schedule with $l_max = 0.05$, $l_min = 0.001$, $T_0 = 10$, $T_mul = 2$ (Loshchilov & Hutter, 2017). Each architecture search is run for 150 epochs. $w$ are initialized with He initialization (He & Sun, 2015). The policy parameters $\theta$ are initialized uniformly in [-0.1, 0.1], and trained with Adam at a learning rate of 0.00035. A tanh constant of 1.10 and a temperature of 2.5 is applied to the controllers logits, and the controller entropy is added to the reward with weight 0.1. For the evaluation, the architecture searched is extended to 17 cells (8 for SPORT8, MIT67 and FLOWERS102), trained for 630 epochs, with batch size 144, and a cosine learning rate schedule with $l_{max} = 0.05$, $l_{min} = 0.001$, $T_0 = 10$, $T_{mul} = 2$.

## A.2 DATASETS

We present here the datasets used and how they are pre-processed.

**CIFAR10.** The CIFAR10 dataset (Krizhevsky, 2009) is a dataset of 10 classes and consists of $50,000$ training images and $10,000$ test images of size $32\times32$.

**CIFAR100.** The CIFAR100 dataset (Krizhevsky, 2009) is a dataset of 100 classes and consists of $50,000$ training images and $10,000$ test images of size $32\times32$.

Each of these datasets is split into a training, validation and testing subsets of size $25,000$, $25,000$ and $10,000$ respectively. For both these datasets, we use standard data pre-processing and augmentation techniques, i.e. subtracting the channel mean and dividing by the channel standard deviation; centrally padding the training images to $40\times40$ and randomly cropping them back to $32\times32$; and randomly clipping them horizontally.

**SPORT8.** This is an action recognition dataset containing 8 sport event categories and a total of 1579 images (Li & Fei-Fei, 2007). The tiny size of this dataset stresses the generalization capabilities of any NAS method applied to it.

**MIT67.** This is a dataset of 67 classes representing different indoor scenes and consists of $15,620$ images of different sizes (Quattoni & Torralba, 2009).

**FLOWERS102.** This is a dataset of 102 classes representing different species of flowers and consists of $8,189$ images of different sizes (Nilsback & Zisserman, 2008).

Each of these datasets is split into a training, validation and testing subsets with proportions $40/40/20$ (%). For each one, we use use standard data pre-processing and augmentation techniques, i.e. subtracting the channel mean and dividing the channel standard deviation, cropping the training images to random size and aspect ratio, resizing them to $224\times224$, and randomly changing their brightness, contrast, and saturation, while resizing test images to $256\times256$ and cropping them at the center.

## A.3 ADDITIONAL RESULTS

### A.3.1 DIFFERENT TRAINING PROTOCOLS FOR RESNET-50

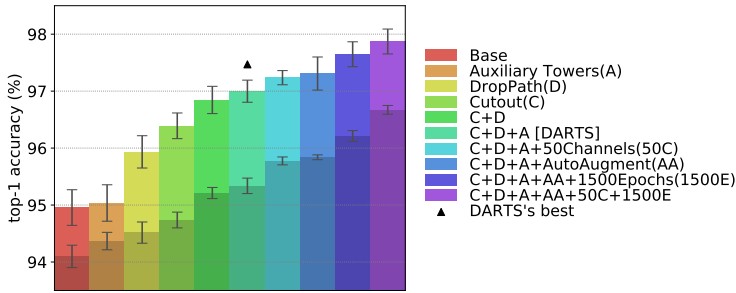

Figure 10: Extension of Figure 3, including results obtained by training a ResNet-50 on CIFAR10. Bars with darker shade are for ResNet-50 and bars with lighter shade are for DARTS (same as Figure 3). Result are for 8 runs of each training protocol. For ResNet-50, the auxiliary tower was added after layer 2. As DropPath (Larsson & Shakhnarovich, 2017) would not have been straightforward to apply, we instead implemented Stocastic Depth (Huang et al., 2016), to a similar effect.

