# OpenReview forum: "NAS evaluation is frustratingly hard"
_ICLR.cc/2020/Conference — Accept (Poster)_

### Official Review · AnonReviewer2 · 2019-10-10
**Official Blind Review #2**

**Rating:** 1

**Review:**

In this submission, the authors conduct a series of experiments on five image classification datasets to compare several existing NAS methods. Based on these experimental results, they point out: 1) how a network is trained (i.e., training protocols/tricks such as DropPath, Cutout) plays an important role for the final accuracy; 2) within the search space, the existing NAS methods perform close to or slightly better than a random sampling baseline; 3) hyperparameters of NAS methods also have significant effect on the performance.

With these interesting findings, I suggest rejecting this submission. The reasons are as follows:
1) For the first finding of training protocol, several existing papers and books already discussed it, such as Li & Talwalkar (2019) and the book chapter {Neural Architecture Search} by Thomas, Jan Hendrik and Frank.

2) For the second finding of the search space and the performance of a randomly sampled architecture, existing work from Facebook AI Research group has studied this. And the existing work gives more experiments and discussion than this submission (from my own perspective). https://arxiv.org/pdf/1904.01569.pdf

3) The conducted experiments in this submission also have certain risks to support the claims/conclusions of it. For example, only datasets of image classification are adopted. Another factor is the hyper-parameter tuning (actually, the authors also mention this in the last paragraph on Page 4). All the compared methods, either NAS methods or random sample baseline, should receive the same training procedure to get a fair experimental comparison.

The above mentioned existing work makes the contributions of this submission less, and the experimental results may not be convincing enough. These lead to a reject.

However, a great point is made by the authors in the last paragraph of Section 6: hyperparameter of NAS methods should be either stable enough or counted toward the cost.

**Experience Assessment:**

I have read many papers in this area.

**Review Assessment: Checking Correctness Of Derivations And Theory:**

N/A

**Review Assessment: Checking Correctness Of Experiments:**

I carefully checked the experiments.

**Review Assessment: Thoroughness In Paper Reading:**

I read the paper thoroughly.

---

> ### Author Response · Authors · 2019-11-06
> **Answer to Reviewer #2 (clarifications)**
>
> We thank the reviewer for the comments but believe there are some key misconceptions.
>
> 1) The reviewer claims that the importance of the training protocol (henceforth 'tricks') has been discussed elsewhere in the literature. This is certainly true, although an evaluation of such tricks is lacking: our paper is the first to systematically perform a study on their impact on the final accuracy.
> While [1] highlights the need to identify the contribution of each component in the NAS pipeline, they do not perform experiments on the importance of different training protocols. Instead, their work focuses on adequate baselines: properly implemented random search (on two datasets). Furthermore, that contribution is fundamentally different from ours, as we focus on random "sampling" as opposed to random "search": we argue that the search spaces are so optimized that there is no need for search at all.
>
> Additionally, in [2] it is stated that "For example, for CIFAR-10, performance substantially improves when using a cosine annealing learning rate schedule (Loshchilov and Hutter, 2017), data augmentation by CutOut (Devries and Taylor, 2017), by MixUp (Zhang et al., 2017) or by a combination of factors (Cubuk et al., 2018), and regularization by Shake-Shake regularization (Gastaldi, 2017) or ScheduledDropPath (Zoph et al., 2018). It is therefore conceivable that improvements in these ingredients have a larger impact on reported performance numbers than the better architectures found by NAS."
> We draw attention to the keyword "conceivable" in the last sentence. The importance of these tricks was suspected but no experiments were performed to support the hypothesis systematically. We provide these experiments and show that the worst architecture with the best training protocol can outperform the best architecture without tricks (Fig. 3, first and last columns).
>
> Both of these references are already acknowledged in our paper and we believe that our contribution substantially complements their message.
>
> 2) The interesting [3] deals with random sampling in a different context. They define a new, graph-based search space and show that randomly sampled architectures from this search space perform well. There is no comparison with searched architectures on their space, nor mention of random sampling in the context of cell-based search spaces (that are currently used by most NAS methods, on which we focus).
> Their approach (tested on a single dataset) showcases the properties of their new search space, while we make a point about multiple existing ones on 5 datasets. Therefore we believe we address a very different issue.
> We have added a reference to it in section 6 (Search space) as an example of broad search space.
>
> 3) We decided to focus on image recognition as that is the topic that is currently receiving more attention in the NAS field. As far as we know, we are the first method to evaluate NAS methods on 5 different benchmarks.
> The different NAS methods have indeed different hyperparameters and training protocols, but the goal here is to evaluate the improvement of each search method over random sampling and not to compare the different methods' absolute performance between each other. Furthermore, keeping hyperparameters and the training protocol of the authors' implementation for each method also ensures we reproduce it with maximum fidelity.
> We would also like to stress that NAS methods and corresponding randomly sampled architectures did receive *exactly* the same training protocol, this being one crucial point of our submission.
>
> [1] Li, Liam, and Ameet Talwalkar. "Random search and reproducibility for neural architecture search." arXiv preprint arXiv:1902.07638 (2019).
>
> [2] T. Elsken, J.H. Metzen, F. Hutter (2019) "Neural Architecture Search: A Survey", Journal of Machine Learning Research 20:1-21.
>
> [3] Xie, Saining, et al. "Exploring randomly wired neural networks for image recognition." arXiv preprint arXiv:1904.01569 (2019).

---

### Official Review · AnonReviewer1 · 2019-10-22
**Official Blind Review #1**

**Rating:** 8

**Review:**


The paper scrutinizes commonly used evaluation strategies for neural architecture search.
The first contribution is to compare architectures found by 5 different search strategies from the literature against randomly sampled architectures from the same underlying search space.
The paper shows that across different image classification datasets, the most neural architecture search methods are not consistently finding solutions that achieve a better performance than random architectures.
The second major contribution of the paper is to show that the state-of-the-art performance achieved by the many neural architecture search methods can be largely attributed to advanced regularization tricks.


In general the paper is well written and easy to follow.
The paper sheds a rather grim light on the current state of neural architecture search, but I think it could raise awareness of common pitfalls and help to make future work more rigorous.
While poorly designed search spaces is maybe a problem that many people in the community are aware of, this is, to the best of my knowledge, the first paper that systematically shows that for several commonly used search spaces there is not much to be optimized.
Besides that, the paper shows that, maybe not surprisingly, the training protocol seems to be more important than the actual architecture search, which I also found rather worrisome.
It would be nice, if Figure 3 could include another bar that shows the performance of a manually designed architecture, for example a residual network, trained with the same regularization techniques.

A caveat of the paper is that mostly local methods with weight-sharing are considered, instead of more global methods, such as evolutionary algorithms or Bayesian optimization, which also showed strong performance on neural architecture search problems in the past.
Furthermore, the paper doesn't mention some recent work in neural architecture search that present more thoroughly designed search spaces, e.g Ying et al.
It would also be helpful if the paper could elaborate, on how better search spaces could be designed.

Nas-bench-101: Towards reproducible neural architecture search
C Ying, A Klein, E Real, E Christiansen, K Murphy, F Hutter
ICML 2019


Minor comments:

- Section 4.1 How are the hyperparameters of drop path probability, cutout length and auxiliary tower weight chosen?



post rebuttal
------------------

I thank the authors for taking the time to answer my questions and performing the additional experiments. The paper highlights two important issues in the current NAS literature: evaluating methods with different search spaces and non-standardised training protocols. I do think that the paper makes an important contribution which hopefully helps future work to over come these flaws.



**Experience Assessment:**

I have published one or two papers in this area.

**Review Assessment: Checking Correctness Of Derivations And Theory:**

N/A

**Review Assessment: Checking Correctness Of Experiments:**

I carefully checked the experiments.

**Review Assessment: Thoroughness In Paper Reading:**

I read the paper thoroughly.

---

> ### Author Response · Authors · 2019-11-06
> **Answer to Reviewer #1**
>
> We thank the reviewer for the positive feedback. We share the concerns regarding NAS but believe these issues can be overcome. We have added in section 6 (Search Space) some notes on how existing search spaces could be improved.
>
> We are currently running experiments with a manually designed ResNet and will add these results when finished.
>
> Regarding Nas-bench-101, we understand it to be a paper presenting a database rather than a specific search strategy and it was unclear for us how to include it in our benchmark (which compares specific search methods to their average architecture).
> However, we have now referenced it in section 6 (Reproducibility), as an example of good practices and highlight that evaluating search methods on it is a reliable way to ensure fairness of comparison in NAS.
>
> Finally, we agree that it would have been valuable to include other methodologies but unfortunately, the search cost of more expensive methods (e.g. evolution) for a repeated evaluation on multiple datasets would have been prohibitive.
>
> Minor:
> We didn't optimize the training hyperparameters (cutout length, drop path probability and auxiliary tower weight) and instead used the values given by the original authors of the different methods. Details of the specific values can be found in Appendix A.1.

---

> > ### Author Response · Authors · 2019-11-11
> > **Added ResNet results**
> >
> > As requested, we have added results with a hand designed ResNet trained with the same training protocols as Fig. 3. This is the new Fig. 10, in the Appendix. Similarly to our previous findings, the manually designed architecture also benefits from the improved training protocols, with the average accuracy increasing from $94.1$ (base) to $96.7$ (all tricks). We have added a reference to this in the main text, at the end of section 4.2.

---

### Official Review · AnonReviewer3 · 2019-10-22
**Official Blind Review #3**

**Rating:** 8

**Review:**

This paper provides a benchmark of 8 the-state-of-the-art NAS methods (DARTS, StacNAS, PDARTS, MANAS, CNAS, NSGANET, ENAS, NAO) on 5 datasets (CIFAR10, CIFAR100, SPORT8, MIT67, FLOWERS102). The paper first points out how fair comparisons of NAS methods is difficult, especially those with different search spaces. The paper proposes making relative comparisons to random "samples" of architectures in search space to remove advantages of expertly engineered search spaces or training protocols. Also, the paper further investigates the case of commonly used the DARTS search space through ablation studies. The results suggest that many sensitive factors such as tricks in evaluation protocols, random seeds, hand-designed macro structures, and depth gaps can have predominant impacts rather than primary factors of NAS such as search strategy. The paper concludes with best practices to mitigate these disturbing factors to design NAS with reproducibility.

This is a very nice paper with extensive empirical evaluations. The topic of empirical comparisons of NAS algorithms is already very difficult to tackle in a fair way, but it gives thought-provoking strategies to evaluate the target NAS algorithms. Also, the fact that even random sampling (without search) provides an incredibly competitive baseline is quite informative and gives a very important recognition on how to design and evaluate NAS.

The paper is well written and well organized, and I have no problems to report, but would like to make sure one thing. In section 4.1, the same 8 initial random architectures from DARTS search space are used for all of the variants? Since the non-negligible impact of random seeds is reported, I just wonder how seeds are controlled in individual experiments.

**Experience Assessment:**

I do not know much about this area.

**Review Assessment: Checking Correctness Of Derivations And Theory:**

N/A

**Review Assessment: Checking Correctness Of Experiments:**

I carefully checked the experiments.

**Review Assessment: Thoroughness In Paper Reading:**

I read the paper thoroughly.

---

> ### Author Response · Authors · 2019-11-06
> **Answer to Reviewer #3**
>
> We thank the reviewer for the positive feedback and enthusiasm.
> We can confirm that in section 4.1, the same 8 random architectures from the DARTS search space were used for all variants in order to provide a fair comparison.

---

### Decision · Program_Chairs · 2019-12-19

**Decision:**

Accept (Poster)

**Comment:**

Summary:
This paper provides comprehensive empirical evidence for some of the systemic issues in the NAS community, for example showing that several published NAS algorithms do not outperform random sampling on previously unseen data and that the training pipeline is more important in the DARTS space than the exact choice of neural architecture. I very much appreciate that code is available for reproducibility.

Reviewer scores and discussion:
The reviewers' scores have very high variance: 2/3 reviewers gave clear acceptance scores (8,8), very much liking the paper, whereas one reviewer gave a clear rejection score (1). In the discussion between the reviewers and the AC, despite the positive comments of the other reviewers, AnonReviewer 2 defended his/her position, arguing that the novelty is too low given previous works. The other reviewers argued against this, emphasizing that it is an important contribution to show empirical evidence for the importance of the training protocol (note that the intended contribution is *not* to introduce these training protocols; they are taken from previous work).

Due to the high variance, I read the paper myself in detail. Here are my own two cents:
- It is not new to compare to a single random sample. Sciuto et al clearly proposed this first; see Figure 1 (c) in https://arxiv.org/abs/1902.08142
- The systematic experiments showing the importance of the training pipeline are very useful, providing proper and much needed empirical evidence for the many existing suggestions that this might be the case. Figure 3 is utterly convincing.
- Throughout, it would be good to put the work into perspective a bit more. E.g., correlations have been studied by many authors before. Also, the paper cites the best practice checklist in the beginning, but does not mention it in the section on best practices (my view is that this paper is in line with that checklist and provides important evidence for several points in it; the checklist also contains other points not being discussed in this paper; it would be good to know whether this paper suggests any new points for the checklist).

Recommendation:
Overall, I firmly believe that this paper is an important contribution to the NAS community. It may be viewed by some as "just" running some experiments, but the experiments it shows are very informative and will impact the community and help guide it in the right direction. I therefore recommend acceptance (as a poster).